

# Strategies for recovery of imbalanced full-scale biogas reactor feeding with palm oil mill effluent

Nantharat Wongfaed[1,*], Prawit Kongjan[2], Wantanasak Suksong[3], Poonsuk Prasertsan[4] and Sompong O-Thong[1,5,*]

[1] Biotechnology Program, Faculty of Science, Thaksin University, Phatthalung, Thailand
[2] Department of Science, Faculty of Science and Technology, Prince of Songkla University, Pattani, Thailand
[3] School of Bioresources and Technology, King Mongkut's University of Technology, Thonburi, Bangkok, Thailand
[4] Research and Development Office, Prince of Songkla University, Songkhla, Thailand
[5] International College, Thaksin University, Songkhla, Thailand
[*] These authors contributed equally to this work.

Corresponding author
Sompong O-Thong,
sompong.o@tsu.ac.th,
sompong.o@gmail.com

## ABSTRACT

**Background**. Full-scale biogas production from palm oil mill effluent (POME) was inhibited by low pH and highly volatile fatty acid (VFA) accumulation. Three strategies were investigated for recovering the anaerobic digestion (AD) imbalance on biogas production, namely the dilution method (tap water vs. biogas effluent), pH adjustment method (NaOH, NaHCO$_3$, Ca(OH)$_2$, oil palm ash), and bioaugmentation (active methane-producing sludge) method. The highly economical and feasible method was selected and validated in a full-scale application.

**Results**. The inhibited sludge from a full-scale biogas reactor could be recovered within 30–36 days by employing various strategies. Dilution of the inhibited sludge with biogas effluent at a ratio of 8:2, pH adjustment with 0.14% w/v NaOH, and 8.0% w/v oil palm ash were considered to be more economically feasible than other strategies tested (dilution with tap water, or pH adjustment with 0.50% w/v Ca(OH)$_2$, or 1.25% NaHCO$_3$ and bioaugmentation) with a recovery time of 30–36 days. The recovered biogas reactor exhibited a 35–83% higher methane yield than self-recovery, with a significantly increased hydrolysis constant (k$_H$) and specific methanogenic activity (SMA). The population of *Clostridium* sp., *Bacillus* sp., and *Methanosarcina* sp. increased in the recovered sludge. The imbalanced full-scale hybrid cover lagoon reactor was recovered within 15 days by dilution with biogas effluent at a ratio of 8:2 and a better result than the lab-scale test (36 days).

**Conclusion**. Dilution of the inhibited sludge with biogas effluent could recover the imbalance of the full-scale POME-biogas reactor with economically feasible and high biogas production performance.

## INTRODUCTION

Palm oil mill effluent (POME) is the main wastewater generated from the palm oil extraction plant which is mostly treated through an anaerobic process with energy production in terms of biogas (*Wu et al., 2010*). Biogas has been identified as one of the most promising renewable technologies based on the socio-economic analysis. The application of biogas production technology to treat POME has expanded in response to demand. Unforeseen process-related accidents occur regularly in biogas plants, where the process is inhibited, and biogas production is reduced. The long-term operation of commercial biogas reactor feeding with POME was confronted with process imbalance by volatile fatty acid (VFA) inhibition, long-chain fatty acids (LCFAs) inhibition, low pH inhibition, and foaming (*Wongfaed, Kongjan & O-Thong, 2015*; *Wongfaed et al., 2020*). The imbalanced biogas reactor resulted in a reduction of biogas production, reduction of chemical oxygen demand (COD) removal efficiency, and failure of the anaerobic digestion (AD) process (*Joo-Hwa & Xiyue, 2000*; *Menardo, Gioelli & Balsari, 2011*). The imbalanced biogas reactor feeding with POME was mainly caused by fluctuations in POME composition and volume that varied depending on the quality of palm fruit, season, harvesting period, and extraction process. The feedstock composition and organic loading rate (OLR) were affected in both bacterial and archaeal communities in the AD process (*Supaphol et al., 2011*; *Xia et al., 2012*). In addition, their fluctuations always caused the process imbalance of high strength feedstock resulting in unstable biogas production performance (*Joo-Hwa & Xiyue, 2000*). Moreover, the overloading of the substrate could inhibit the AD process, resulting in losses of methane yield of up to 30% (*Fotidis et al., 2014*). Reactor acidification by organic overload is one of the most common reasons for this AD process imbalance (*Akuzawa et al., 2011*) due to the rapid accumulation of VFA from uncoupling between the acid producers and consumers. The consequences of the AD process imbalance are financial losses due to reduced biogas yield and increased staff deployment and chemical addition cost. Therefore, it is necessary to solve these problems on time.

The typical recovery strategy for the AD process imbalance is stopped feeding to restore the ecological function of microorganisms in the AD system via self-recovery. However, this strategy requires a long time and is not economically feasible. The stop feeding strategy, combined with the addition of trace elements, could accelerate the recovery process of the inhibited AD reactor but still requires a long stop feeding period (*Voelklein et al., 2017*). The recovery of VFA and low pH inhibition in real-time without stop feeding is still a significant challenge. Adjusting the pH of the inhibited AD reactor to near-neutral was often applied to enhance the buffering capacity of the AD system against VFA disturbance, with low cost and smooth operation. The adjusted pH in the AD reactor could recover the AD process from low pH inhibition with a stable operation (*Zhang, Wang & Jiang, 2013*). Alkaline addition to the AD reactor improved the buffering capacity to meet the requirements of the microbial populations (*Zhang et al., 2016*) and enhance activities of the acidogenic bacteria and methanogenic archaea (*Zhang, Qiu & Chen, 2012*). Alkaline substances, such as $Na_2CO_3$ and $NaHCO_3$, exhibited more pronounced effects on the stability of the AD process than NaOH due to $CO_3^{2-}$ and $HCO_3^-$ having a higher buffering capacity than $OH^-$

(*Jun et al., 2009*). Additionally, instead of alkaline chemicals, wood ash was used to adjust the pH of the AD process as a cheap material alternative (*Saritpongteeraka & Chaiprapat, 2008*). Oil palm ash was used to adjust the pH of POME with high biogas production, rather than raw POME, due to the releasing buffer capacity and micronutrient (*Gómez et al., 2006*). However, the pH adjustment strategy could not recover the imbalanced AD reactor, but only delayed the AD process failure (*Gómez et al., 2006*). Nevertheless, the addition of fresh and active methane-producing sludge, with the addition of micronutrient, has been used to recover the imbalanced AD reactor (*Lee & Shoda, 2008*; *Qiang et al., 2013*). The re-inoculation (*Wu et al., 2015*) or bioaugmentation (*Li et al., 2018*) of high-activity anaerobic microorganism was used to restart the out-of-order AD reactor, and this was significantly effective in the short-term, although expensive. The combination of the pH adjustment with trace elements and re-inoculation was always useful but costly (*Zhang, Xing & Li, 2018*). All strategies described above have proved to be effective methods to recover the imbalanced AD process. However, a systematic and comprehensive evaluation of the recovery strategies from the imbalanced AD reactor feed with POME has yet to be reported.

This work aims to recover the imbalanced AD reactor feed with POME by pH adjustment with an alkaline substance, diluting the toxic compounds with tap water and biogas effluent, and re-inoculation addition of active methane-producing sludge. The microbial community responsible for each recovery strategy was investigated, and the knowledge from our research can provide economically feasible and rapid recovery methods for imbalanced commercial biogas reactors.

## MATERIALS & METHODS

### Characteristics of inhibited AD sludge, POME, biogas effluent, and active methane-producing sludge

An inhibited sludge sample was collected from the mesophilic biogas plant ($40 \pm 2$ °C) of a palm oil mill, Prasang Green Power Co., Ltd., Surat Thani Province, Thailand. The biogas reactor was operated in continuous mode, feeding with POME at a high OLR (4.5 g-COD/L/d), resulting in acidification of the AD reactor. Biogas effluent, POME, and active methane-producing sludge were collected from the biogas plant at Pitak Palm Oil Co., Ltd., Trang Province, Thailand, and analyzed for their characteristics, according to the procedure described in *APHA (2012)*. The inhibited sludge had an acidic pH (3.9), 4.8 g/L of total volatile fatty acids (tVFA), 17.0 g/L of suspended solids (SS), and 14.5 g/L of volatile suspended solids (VSS). The active methane-producing sludge had neutral pH (7.5), very low tVFA (0.92 g/L), but a high SS and VSS content of 59.8 g/L and 52.2 g/L, respectively.

### Recovery of AD process imbalance

Experiments carried out in a batch reactor, an AD process imbalance, indicated by inhibited sludge, were recovered using three strategies. Firstly, the inhibited sludge sample was diluted with tap water (TW) and biogas effluent (BE) at a ratio of 9:1, 8:2, 7:3, 6:4, and 5:5, respectively, as a dilution strategy. Secondly, the pH of inhibited sludge was adjusted

using 0.85–1.50% w/v sodium hydrogen carbonate (NaHCO$_3$), 0.10–0.14% w/v sodium hydroxide (NaOH), 0.10–0.50% w/v calcium hydroxide Ca(OH)$_2$ and 6.0–10.0% w/v oil palm ash as pH adjustment strategy. Thirdly, the inhibited sludge was recovered by adding active methane-producing sludge at 5, 10, 15, 20, 25, 30, 35, 40, 45, and 50% v/v called re-inoculation or bioaugmentation strategy. All strategies were combined with 20% v/v POME addition as low flow rate feeding (40 m$^3$-POME/d), instead of stop feeding, and self-recovery was used as a control. All strategies were tested at an inhibited sludge sample concentration of 1 g VSS. All experiments were flushed with N$_2$: CO$_2$ mixed at 80:20 ratios to create the anaerobic condition and secured tightly with a butyl-rubber septum and aluminum cap. The experiment was carried out in triplicate and at a temperature of (40 ± 2°C) for 45 days. The biogas production in the headspace was measured via the water displacement method, and the biogas content was analyzed by a gas chromatograph equipped with thermal conductivity detectors (GC-TCD). Microbial sludge from each treatment was analyzed for the microbial community structure using polymerase chain reaction denaturing gradient gel electrophoresis (PCR-DGGE) techniques. The specific methanogenic activity (SMA) of the inhibited sludge and recovery sludge was also determined (*Hussain & Dubey, 2017*). The recovery time is the incubation time for methane production reaches 90% of maximum methane production in the recovery experiment and simulation results (*Wu et al., 2015*).

## The validity of lab-scale results in the full-scale recovery of AD process imbalance

The inhibited sludge from a 6,000 m$^3$ full-scale hybrid cover lagoon reactor was tested in the lab-scale experiment for recovery strategies. The selected strategy (dilution with biogas effluent) was applied to a 6,000 m$^3$ full-scale hybrid cover lagoon reactor (Prasang Green Power Co., Ltd., Surat Thani Province, Thailand) for recovery of the AD process imbalance and validated the lab-scale results. The biogas reactor was operated at hydraulic retention times (HRT) of 30 days. The organic loading rate (OLR) was reduced from 4.5 kg COD/m$^3$/d to 1.25 kg COD/m$^3$/d. The 50 m$^3$ of inhibited sludge and 10 m$^3$ of POME were diluted with biogas effluent at a ratio of 8:2 every day for two weeks before adding to the full-scale hybrid cover lagoon reactor. The reactor was operated normally when the pH of inhibited sludge was increased to 7.5. Biogas effluent with pH 7.8 was collected from another biogas reactor effluent with good performance at Prasang Green Power Co., Ltd., Surat Thani Province, Thailand. POME was collected from the palm oil mill at the biogas plant at Prasang Green Power Co., Ltd., Surat Thani Province, Thailand. The pH and methane production rate of full-scale biogas sludge were monitored daily. The SMA of full-scale biogas sludge was monitored every 3 days. Acetate was used as a substrate during SMA tests as representative acetoclastic bacteria to investigate the reactor performance during the recovery process.

## Microbial activity and microbial community analysis

The inhibited and recovered sludge methanogenic activities were evaluated by the SMA test using avicel (cellulose), glucose, gelatin, and acetic acid as a representative microbial population group of hydrolytic bacteria, acidogenic bacteria, proteolytic

bacteria and, methanogenic archaea, respectively. This evaluates the anaerobic sludge capability to convert an organic substrate into methane, escaping quickly to the gas phase, reducing the COD in a liquid phase. The SMA value was calculated by the slope of methane production (based on g COD of $CH_4$) against incubation time and divided with VSS added of sludge sample (*Hussain & Dubey, 2017*). The microbial community structure was analyzed by polymerase chain reaction denaturing gradient gel electrophoresis (PCR-DGGE) techniques, according to *Prasertsan, O-Thong & Birkeland (2009)*. The 0.2 g of the sludge sample was extracted for genomic DNA using the Ultraclean Soil DNA Kit (MoBio Laboratory Inc., USA). The 16S rDNA gene of bacteria was amplified by the first PCR with universal primer 27f (GAGTTTGATCCTTGGCTCAG) and 1525r (AAGGAGGTGWTCCARCC). 16S rDNA gene for archaea was amplified using Arch21f primers (TTCCGGGTTGATCCYGCCGGA) and Arch958r (YCCGGCGTTGAMTCCAATT). The V3 region of bacteria was amplified in a second PCR by primer 357f (CTCCTACGGGAGGCAGCAG) with CG clamp and 518r (GTATTACCGCGGCTGCTGG), using the first bacteria PCR as a template. The V3 region of archaea was amplified in a second PCR by primer 340f (CCTACGGG-GYGCASCAG) with CG clamp and 519r (TTACCGCGGCKGCTG), using the product of first archaea PCR as a template. Second PCR products performed the denaturing gradient gel electrophoresis (DGGE) analysis with 6% polyacrylamide gel for bacteria and 8%polyacrylamide gel for archaea containing a linear of urea/formamide gradient with denaturant ranging from 40% to 70% in 0.5 TAE buffer at 20 volts for 20 min, and 70 volts for 15 h, at a constant temperature of 60 °C. Sybr-Gold was stained in the DGGE gels for 60 min and photographed on the Gel Doc XR system (Bio-Rad Laboratories). Predominant DGGE bands were excised with a sterile tip and suspended in 30 μL sterilized Milli-Q water. Excised DGGE band was incubated at 4 °C overnight and re-amplified by PCR using the same primers without the GC clamp. PCR products were purified and sequenced by Macrogen Inc. (Seoul, South Korea). Closest matches for partial 16S rRNA gene sequences were identified by database searches in Gene Bank using BLAST (*Tatusova et al., 2016*).

## Analytical methods and calculation

The biogas composition ($H_2$, $N_2$, $CH_4$, and $CO_2$) was determined by gas chromatograph GC-8A (Shimadzu, Kyoto, Japan) with a 1-meter stainless steel column packed with Shin Carbon (60/80 mesh) equipped with thermal conductivity detectors (TCD). The argon at a flow rate of 14 mL/min was used as the carrier gas. The temperatures of the oven, detector, and injection port were at 40 °C, 100 °C, and 120 °C, respectively. The 0.5 mL of the biogas sample was injected in duplicate for each reactor. The daily biogas production for each reactor was counted using the water displacement method (*Yan et al., 2015*). The chemical and physical composition of POME, biogas effluent, active methane-producing sludge, and inhibited sludge were determined for pH, lipid content, total solids (TS), volatile solids (VS), volatile suspended solids (VSS), total nitrogen (TN), total volatile fatty acid (tVFA), and alkalinity according to Standard Methods for the Examination of Water and Wastewater (*APHA, 2012*). Determination of TS was performed at a temperature of 90 °C instead of 105 °C, till constant weight to avoid decreasing of VFAs (*Angelidaki et*

*al., 2009*). The VFAs composition was determined through a gas chromatograph GC-17A (Shimadzu, Kyoto, Japan) with a Stabilwax®-DA fused silica column (30 m of length, 0.53 mm of diameter, 85 °C) connected to a flame ionization detector (FID) at 240 °C. The helium at 30 mL/min was used as the carrier gas. The VFA samples were collected by syringe (one mL) and filtered through a nylon membrane (0.2 µm). The filtrated samples were acidified to pH 3.0–3.2 with 30% (v/v) phosphoric acid for VFAs analysis *Raposo et al. (2015)*. Buswell's equation was used to calculate theoretical methane yield, assuming the total stoichiometry conversion of the organic matter to methane and carbon dioxide (*Buswell & Mueller, 1952*). The hydrolysis constants ($k_H$) were determined by using the first-order kinetic model in Eq. (1) according to the protocol of (*Raposo et al., 2006*). Where B(t) is the cumulative methane yield (mL-$CH_4$/g-$VS_{added}$) at time t, and $B_\infty$ is the maximum cumulative methane yield (mL-$CH_4$/g-$VS_{added}$). $k_H$ is the hydrolysis constant ($d^{-1}$); t is the fermentation time; λ is the lag phase (day).

$$B(t) = B_\infty[1 - \exp(-k_H(t - \lambda))]. \qquad (1)$$

The modified Gompertz model was used to predict the methane production, methane production rate, and lag phase (*Nopharatana, Pullammanappallil & Clarke, 2007*) as follows in Eq. (2).

$$M = P.\exp\{-\exp[\frac{Rme}{p}(\lambda - t) + 1]\} \qquad (2)$$

Where M is the cumulative methane yield at the time t (mL-$CH_4$/g-$VS_{added}$); P is the maximum cumulative methane yield (mL-$CH_4$/g-$VS_{added}$); Rm is the maximum methane production rate (mL-$CH_4$/g-VS/d); λ is the lag phase (d); t is the fermentation time, and e is the Euler constant (2.718282). The economic evaluation of each recovery strategy was calculated from the chemical addition cost, human resources, energy consumption, biogas loss, and biogas production during the recovery period according to the current market price and the average price of industrial water in Thailand. Biogas price is 0.21 USD/$m^3$. The cost of NaOH, Ca$(OH)_2$, and $NaHCO_3$ was 0.35, 0.2, and 0.3 USD/kg, respectively. The cost of human resources was 10 USD/people/d. The cost of electricity consumption was 0.13 USD/kWh. Biogas loss is the average daily biogas during a good performance period (3,000 $m^3$/d) minus biogas production during the recovery period.

## RESULTS

### Recovery of AD process imbalance

The AD process imbalance is caused by a low pH and high VFAs accumulation due to organic overload and low degradation efficiency. Inhibited sludge had low pH (3.9) and a high total VFA (4.8 g/L) with butyric acid (2.5 g/L) and acetic acid (1.8 g/L) as the main VFA (Table 1). It also had a low SS and VSS of 14.5 g/L and 11.0 g/L, respectively, indicating a low number of anaerobic microorganisms in the systems. The specific methanogenic activity (SMA) of the inhibited sludge with glucose, acetic acid, avicel (cellulose), and gelatin was 0.208, 0.344, 0.401, and 0.065 g$CH_4$-COD/gVSS/d, respectively (Fig. 1). The SMA of acetic acid (0.344 g-$CH_4$-COD/g-VSS/d) was lower than the SMA of avicel

**Table 1  The characteristics of inhibited sludge, active methane-producing sludge, POME, and biogas effluent.**

| Parameter | Unit | POME | Biogas effluent | Active methane-producing sludge | Inhibited sludge |
|---|---|---|---|---|---|
| pH | – | $4.1 \pm 0.1$ | $7.8 \pm 0.1$ | $7.5 \pm 0.1$ | $3.9 \pm 0.1$ |
| Total solids (TS) | g/L | $55.4 \pm 0.2$ | $13.4 \pm 0.3$ | $68.4 \pm 0.3$ | $24.0 \pm 0.2$ |
| Volatile solids (VS) | g/L | $45.1 \pm 0.3$ | $4.8 \pm 0.2$ | $61.5 \pm 0.2$ | $17.0 \pm 0.1$ |
| Suspended solids (SS) | g/L | $34 \pm 0.3$ | $2.4 \pm 0.3$ | $59.8 \pm 0.1$ | $14.5 \pm 0.2$ |
| Volatile suspended solids (VSS) | g/L | $17.0 \pm 1.2$ | $1.3 \pm 0.4$ | $52.2 \pm 0.3$ | $11.0 \pm 0.3$ |
| Total nitrogen (TN) | g/L | $1.2 \pm 0.2$ | $0.2 \pm 0.3$ | $2.5 \pm 0.4$ | $0.7 \pm 0.2$ |
| Alkalinity | g/L as $CaCO_3$ | $2.9 \pm 0.2$ | $3.9 \pm 0.2$ | $6.1 \pm 0.1$ | $2.4 \pm 0.2$ |
| Lipid | g/L | $6.5 \pm 0.3$ | $0.1 \pm 0.02$ | $0.5 \pm 0.1$ | $4.4 \pm 0.1$ |
| Total chemical oxygen demand (tCOD) | g/L | $59.0 \pm 0.1$ | $7.5 \pm 0.1$ | N.D. | $28.9 \pm 0.2$ |
| Soluble chemical oxygen demand (sCOD) | g/L | $38.2 \pm 0.3$ | $1.2 \pm 0.3$ | $2.5 \pm 0.2$ | $23.9 \pm 0.3$ |
| Total volatile fatty acids (TVFAs) | g/L | $1.3 \pm 0.1$ | $0.2 \pm 0.2$ | $0.9 \pm 0.3$ | $4.8 \pm 0.1$ |
| Acetic acid | g/L | $0.4 \pm 0.04$ | $0.02 \pm 0.01$ | $0.3 \pm 0.01$ | $1.8 \pm 0.2$ |
| Propionic acid | g/L | $0.06 \pm 0.01$ | $0.05 \pm 0.02$ | $0.07 \pm 0.02$ | $1.4 \pm 0.1$ |
| Isobutyric acid | g/L | $0.03 \pm 0.01$ | $0.03 \pm 0.01$ | $0.05 \pm 0.01$ | $0.3 \pm 0.1$ |
| Butyric acid | g/L | $0.6 \pm 0.05$ | $0.02 \pm 0.01$ | $0.5 \pm 0.04$ | $2.5 \pm 0.1$ |

**Notes.**
 N.D., Not determined.

(cellulose), indicating a low number of methanogenic archaea for reducing acetate to $CH_4$ and $CO_2$ but a high number of hydrolytic bacteria and acidogenic bacteria for acid production. The pH of inhibited sludge was gradually increased from 5.7 to 7.8. Data files regarding SMA analysis of inhibited microbial sludge, self-recovery sludge, and all recovered strategy were showed in Data S1. The acidic pH directly affected microbial activity, leading to a low methane production rate (40.8 mL-$CH_4$/d) and extended recovery time (49.4 days). The self-recovery had low hydrolysis constant ($k_H$) (0.005 $d^{-1}$) with a long lag phase (21.2 days) and low methane yield (209 mL-$CH_4$/g-$VS_{added}$) were observed in self-recovery (Table 2). The specific methanogenic activity of self-recovery was lower than those of the inhibited sludge for cellulose and glucose as substrate (0.112 and 0.106 g$CH_4$-COD/gVSS/d, respectively) but higher for acetic acids and gelatin substrate (0.481 and 0.122 g$CH_4$-COD/gVSS/d, respectively) (Fig. 1). The SMA test using avicel (cellulose), glucose, gelatin, and acetic acid as a substrate is a representative microbial population group of hydrolytic bacteria, acidogenic bacteria proteolytic bacteria, and methanogenic archaea, respectively. Methane production was achieved from self-recovery but the low activity of specific methanogenic activity, indicating low active methane-producing microorganisms in the self-recovery strategy. A more detailed biogas production data of each recovery strategy and self- recovery will be provided in Data S2.

The recovery by dilution strategy with biogas effluent (BE) gave higher methane yield (214–282 mL-$CH_4$/g-$VS_{added}$) and methane production rate (53.4–108.0 mL-$CH_4$/d) than the recovery by dilution with tap water (TW) (177-190 mL-$CH_4$/g-$VS_{added}$ and 40.7–62.5 mL-$CH_4$/d, respectively) and the self-recovery (Table 2). The lag phase of recovery by dilution with BE (7.1–12.3 days) and dilution with TW (9.9–15.1 days) were significantly

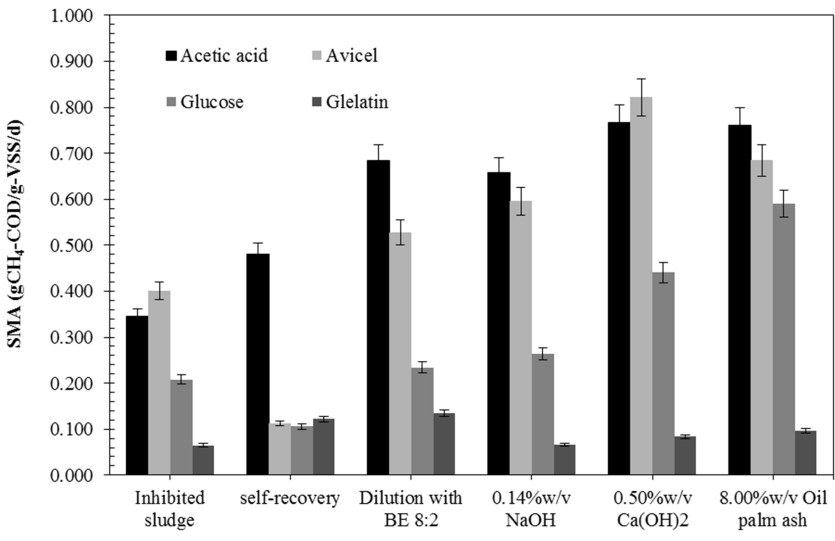

**Figure 1** Specific methanogenic activity (SMA) of the inhibited sludge and recovered sludge by different recovery strategies.

shorter than self-recovery (21.2 days). The recovery time of dilution with TW and BE was in the same range (33.2–41.7 days and 32.8–42.0 days, respectively). The dilution with TW could reduce toxicity in the inhibited sludge so that lag phase and recovery time could be decreased, but not enhanced methane-producing microorganisms resulting in low methane yield and methane production rate. The recovery by dilution with BE at a ratio of 8:2 to 5:5 had a 20–32% shorter lag phase and recovery time than the self-recovery. The recovery by dilution with BE could enhance active methane-producing microorganisms in sludge and resulted in a 2.2 fold increase of the methane production rate (91.4–92.8 mL-CH$_4$/d compared to 40.8 mL-CH$_4$/d, respectively). The recovery by dilution with BE had specific methanogenic activity higher than self-recovery. The specific methanogenic activity of recovery sludge with glucose, acetic acid, cellulose, and gelatin was 0.234, 0.684, 0.528, and 0.134 gCH$_4$-COD/gVSS/d, respectively. The methane yield of dilution with BE strategy was 34.9% higher than the self-recovery and showed better growth of methanogens under suitable pH (6.5–7.3). Therefore, the dilution of the inhibited sludge with BE at a ratio of 8:2 was a suitable strategy to accelerate the recovery process of the sludge from the inhibited state.

The recovery by adjusting pH strategy with all concentrations tested of (0.10–0.5% w/v) Ca(OH)$_2$, (0.10–0.14% w/v) NaOH, (6.00–10.00% w/v) oil palm ash, except (0.85–1.25% w/v NaHCO$_3$) exhibited higher methane yield and methane production rate than the self-recovery (Table 2). The highest methane yield (383 mL-CH$_4$/g-VS$_{added}$) was achieved from the recovery by adjusting pH with 0.14% w/v NaOH with the methane production rate of 111.7 mL-CH$_4$/d. On the other hand, the highest methane production rate (226.3 mL-CH$_4$/d) was achieved from the recovery by adjusting pH with 8.0% w/v oil palm ash with the hydrolysis constant (k$_H$) of 0.007 d$^{-1}$ and lag phase of 8.2 days with the recovery time of 32.4 days. In terms of the specific SMA of recovery sludge by three

**Table 2  Performance of self-recovery and recovered sludge by various strategies, with the colors ranging from the high performances in dark red to low performance in light red.**

| Strategies | Initial pH | Final pH | Methane yield (mL-CH$_4$/ g-VS$_{added}$) | Methane Production rate (mL-CH$_4$/d) | $k_H$ (d$^{-1}$) | Lag phase (d) | Recovery time (d) |
|---|---|---|---|---|---|---|---|
| Self-recovery | 5.7 | 7.8 | 209 | 40.8 | 0.005 | 21.2 | 49.4 |
| Dilution with TW 9:1 | 6.2 | 7.3 | 190 | 42.7 | 0.007 | 15.1 | 41.7 |
| Dilution with TW 8:2 | 6.3 | 7.4 | 187 | 40.6 | 0.006 | 12.4 | 39.2 |
| Dilution with TW 7:3 | 6.3 | 7.4 | 178 | 46.0 | 0.007 | 14.3 | 38.1 |
| Dilution with TW 6:4 | 6.5 | 7.5 | 173 | 46.4 | 0.007 | 10.9 | 36.5 |
| Dilution with TW 5:5 | 6.5 | 7.5 | 177 | 62.5 | 0.006 | 9.9 | 33.2 |
| Dilution with BE 9:1 | 6.7 | 7.3 | 214 | 53.4 | 0.008 | 12.3 | 42.0 |
| Dilution with BE 8:2 | 6.8 | 7.4 | 282 | 91.4 | 0.009 | 7.1 | 36.4 |
| Dilution with BE 7:3 | 6.9 | 7.5 | 253 | 108.0 | 0.008 | 7.7 | 30.8 |
| Dilution with BE 6:4 | 6.5 | 7.6 | 229 | 100.4 | 0.009 | 7.2 | 31.2 |
| Dilution with BE 5:5 | 6.5 | 7.8 | 230 | 92.8 | 0.008 | 9.9 | 32.8 |
| 0.10%w/v NaOH | 6.5 | 8.6 | 218 | 85.9 | 0.007 | 9.5 | 30.8 |
| 0.11%w/v NaOH | 6.6 | 8.8 | 238 | 87.3 | 0.007 | 9.5 | 30.0 |
| 0.12%w/v NaOH | 6.8 | 8.8 | 278 | 87.0 | 0.006 | 8.9 | 35.0 |
| 0.13%w/v NaOH | 7.0 | 8.2 | 263 | 85.9 | 0.006 | 9.1 | 34.1 |
| 0.14%w/v NaOH | 7.3 | 8.5 | 383 | 111.7 | 0.006 | 9.0 | 35.0 |
| 0.85%w/v NaHCO$_3$ | 6.7 | 7.2 | 223 | 31.8 | 0.004 | 14.0 | 45.8 |
| 1.00%w/v NaHCO$_3$ | 6.8 | 7.3 | 224 | 24.2 | 0.004 | 14.0 | 58.8 |
| 1.25%w/v NaHCO$_3$ | 6.9 | 7.5 | 268 | 27.2 | 0.005 | 14.0 | 61.6 |
| 1.45%w/v NaHCO$_3$ | 6.9 | 7.6 | 168 | 20.8 | 0.004 | 16.0 | 58.5 |
| 1.50%w/v NaHCO$_3$ | 7.0 | 7.7 | 158 | 22.0 | 0.004 | 16.0 | 53.4 |
| 0.10%w/v Ca(OH)$_2$ | 6.7 | 8.1 | 319 | 103.8 | 0.008 | 7.6 | 36.7 |
| 0.20%w/v Ca(OH)$_2$ | 6.8 | 8.1 | 333 | 129.5 | 0.007 | 9.0 | 35.3 |
| 0.30%w/v Ca(OH)$_2$ | 6.8 | 8.0 | 368 | 112.0 | 0.007 | 8.3 | 35.3 |
| 0.40%w/v Ca(OH)$_2$ | 7.0 | 8.3 | 373 | 151.4 | 0.008 | 9.1 | 33.8 |
| 0.50%w/v Ca(OH)$_2$ | 7.1 | 8.6 | 360 | 137.7 | 0.006 | 9.4 | 33.9 |
| 6.00%w/v Oil palm ash | 6.5 | 7.5 | 239 | 155.4 | 0.007 | 7.9 | 33.2 |
| 7.00%w/v Oil palm ash | 6.9 | 7.8 | 265 | 155.2 | 0.006 | 8.6 | 32.1 |
| 8.00%w/v Oil palm ash | 6.9 | 8.0 | 347 | 226.3 | 0.007 | 8.2 | 32.4 |
| 9.00%w/v Oil palm ash | 6.9 | 8.3 | 218 | 147.5 | 0.006 | 9.2 | 33.0 |
| 10.00%w/v Oil palm ash | 6.9 | 8.7 | 211 | 152.2 | 0.006 | 9.1 | 32.5 |

alkaline sources for adjusting pH, Ca(OH)$_2$ showed the highest SMA on acetic acid and cellulose, as substrates were 0.767 and 0.821 gCH$_4$-COD/gVSS/d, respectively. In contrast, the highest SMA on glucose and gelatin (0.591 and 0.096 gCH$_4$-COD/gVSS/d, respectively) were obtained from recovery by adjusting pH with 8.00% w/v oil palm ash. Thus, the pH adjustment strategy using 0.14% w/v NaOH and 0.40% w/v Ca(OH)$_2$ gave the highest methane yields of 383 and 373 mL-CH$_4$/g.VS$_{added}$, respectively, while using 8.00% w/v oil palm ash and 0.40% w/v Ca(OH)$_2$ gave the highest methane production rates of 226.3 and 151.4 mL-CH$_4$/d, respectively. The 0.40% w/v Ca(OH)$_2$ was demonstrated effective

**Table 3  Performance of recovered sludge by addition of active methane-producing sludge.** With the colors ranging from the high performances in dark red to low performance in light red.

| Active methane-producing sludge(% v/v) | InitialpH | Final pH | Methane yield (mL-CH$_4$/g-VS$_{added}$) | Methane production rate (mL-CH$_4$/d) | $k_H$(d$^{-1}$) | Lag phase (d) | Recovery time (d) |
|---|---|---|---|---|---|---|---|
| 5 | 6.4 | 7.5 | 212 | 40.4 | 0.007 | 15.2 | 45.8 |
| 10 | 6.4 | 7.5 | 216 | 47.2 | 0.007 | 15.1 | 46.1 |
| 15 | 6.4 | 7.5 | 214 | 60.8 | 0.007 | 15.0 | 47.2 |
| 20 | 6.4 | 7.5 | 222 | 77.0 | 0.006 | 15.0 | 45.2 |
| 25 | 6.4 | 7.5 | 222 | 70.7 | 0.006 | 15.0 | 44.2 |
| 30 | 6.4 | 7.6 | 224 | 65.9 | 0.006 | 13.2 | 43.6 |
| 35 | 6.4 | 7.6 | 222 | 67.9 | 0.006 | 12.0 | 43.9 |
| 40 | 6.4 | 7.6 | 227 | 80.9 | 0.006 | 12.0 | 44.0 |
| 45 | 6.4 | 7.9 | 230 | 80.1 | 0.007 | 11.0 | 44.5 |
| 50 | 6.4 | 8.0 | 237 | 83.5 | 0.008 | 11.0 | 45.8 |

in accelerating recovery of the inhibited sludge by saving 31.6% recovery time (from 49.4 days to 33.8 days) and enhancing the methane yield of 78.5% (from 209 to 373 mL-CH$_4$/g.VS$_{added}$) when compared with self-recovery. Nevertheless, the pH adjustment strategy from the best results of each source of alkaline (0.14% w/v NaOH, 0.40% w/v Ca(OH)$_2$, 1.25% w/v NaHCO$_3$, and 8.00% w/v oil palm ash) was selected for economic evaluation compared to the dilution strategy (dilution with BE at 8:2 ratio).

The recovery by bioaugmentation or re-inoculation of active methane-producing sludge at 5–50% into the inhibited sludge was able to accelerate the recovery process in terms of methane yield (212-237 mL-CH$_4$/g-VS$_{added}$), methane production rate (40.4–83.5 mL-CH$_4$/d), lag phase (15.2–11.0 days) but not reduced the recovery time (43.6–47.2 days) (Table 3). Its $k_H$ value was 0.006–0.008 d$^{-1}$. The addition of active methane-producing sludge at 15–50% had a methane production rate higher than the self-recovery because it can improve the amount of active biomass. The recovery of this strategy gave a small improvement with a long recovery time of 45.8 days and not difference with self- recovery (49.20 days). In particular, $k_H$ of the addition of active methane-producing sludge strategy was small increased when compared with self-recovery. The amount of active methane-producing sludge at 30%–50% was suitable for recovery of the inhibited sludge with a shorter lag phase and increased tolerance of microorganism to low pH and high VFAs. However, the addition of active methane-producing sludge strategy had lower recovery efficiency than alkali addition due to low buffering capacity.

## Economic evaluation

The energy and economic evaluation of each recovery strategy were provided in this study. The price of chemical addition, energy consumption, biogas loss, and human resources was achieved according to the current market price and the average price of industrial water in Thailand. According to Table 4, there were no extra-economic benefits from either of the recovery strategies. The recovery by NaOH addition, dilution with biogas effluent, and oil palm ash addition could reduce the recovery cost of the inhibited AD systems more than other strategies. The net profit of strategies for recovery inhibited sludge is in

**Table 4  The economic evaluation of different strategies for the recovery of the imbalanced AD reactor.**

| Items | Dilution with tap water (TW) | Dilution with biogas effluent (BE) | NaOH addition | Ca(OH)$_2$ addition | NaHCO$_3$ addition | Oil palm ash addition |
|---|---|---|---|---|---|---|
| Chemical addition cost (USD/m$^3$/d) | −0.02 | 0.00 | −0.49 | −0.19 | −1.73 | 0.00 |
| Human resource cost (USD/m$^3$/d) | −0.05 | −0.05 | −0.15 | −0.15 | −0.15 | −0.15 |
| Energy consumption (USD/m$^3$/d) | −0.10 | −0.10 | −0.02 | −0.02 | −0.02 | −0.02 |
| Biogas loss (USD/m$^3$/d) | −4.26 | −3.46 | −3.20 | −3.73 | −5.30 | −3.46 |
| Biogas production (USD/m$^3$/d) | 1.04 | 1.84 | 2.10 | 1.60 | 0.00 | 1.84 |
| Net profit (USD/m$^3$/d) | −3.39 | −1.77 | −1.76 | −2.49 | −7.20 | −1.79 |

the following order (USD/m$^3$/d): NaOH addition (−1.76), dilution with biogas effluent (−1.77), oil palm ash addition (−1.79) Ca(OH)$_2$ addition (−2.49), dilution with tap water (−3.33), and NaHCO$_3$ addition (−7.2). The recovery by 0.14% w/v NaOH addition corresponding to the addition of NaOH of 1.40 kg/m$^3$-inhibited sludge had the chemical addition cost of 0.49 USD/m$^3$/d. The recovery by dilution with biogas effluent at a ratio of 8:2 corresponding to 0.2 m$^3$/m$^3$-inhibited had no cost for biogas effluent. The recovery by 8.00% w/v of oil palm ash addition corresponding oil palm ash of 80 kg/m$^3$-inhibited sludge had no oil palm ash cost. Oil palm ash is a by-product obtained by burning fibers, shells, and empty fruit bunches as fuel in palm oil mill boilers, while biogas effluent is that from AD digester free of charge. Thus, we suggest that NaOH addition, dilution with biogas effluent, and oil palm ash addition could be economically feasible strategies to recover the inhibited AD system feeding with POME.

## The validity of lab-scale results in the full-scale recovery of AD process imbalance

Full-scale recovery was conducted by diluting 50 m$^3$ of inhibited sludge with biogas effluent at a ratio of 8:2, which was added to 6,000 m$^3$ biogas reactors every day. The AD imbalance reactor was maintained by a low feeding rate of 10 m$^3$-POME/d. After dilution, the pH increased from 5.6 to 6.8 in the first week and 7.8 in the second week. After that, the reactor was operated normally with a feeding rate of 200 m$^3$-POME/d. The methane production rates of the first, second, and third weeks of dilution with biogas effluent were 0.8, 2.0, and 2.86 m$^3$-CH$_4$/m$^3$-reactor /d with the pH values of 6.8, 7.8, and 7.8, respectively (Fig. 2). The SMA values of the first, second, and third weeks of dilution with biogas effluent were also increased to 0.44, 0.70, and 0.71 g-CH$_4$-COD/g-VSS/d, respectively. Results indicated that the reactor recovered within two weeks after dilution with biogas effluent. Therefore, this recovery time (15 days) gave a better result than lab-scale reactor (36.4 days).

## Microbial community responsible for high potential recovery strategies

The microbial community from the four high potential recovery strategies (dilution with BE at a ratio of 8:2, 0.14% w/v NaOH addition, 0.50% w/v Ca(OH)$_2$ addition, and 8.00% w/v oil palm ash addition) indicated by short time lag phase, short recovery time, high SMA activity, and high methane production was analyzed. The heat map of the bacterial

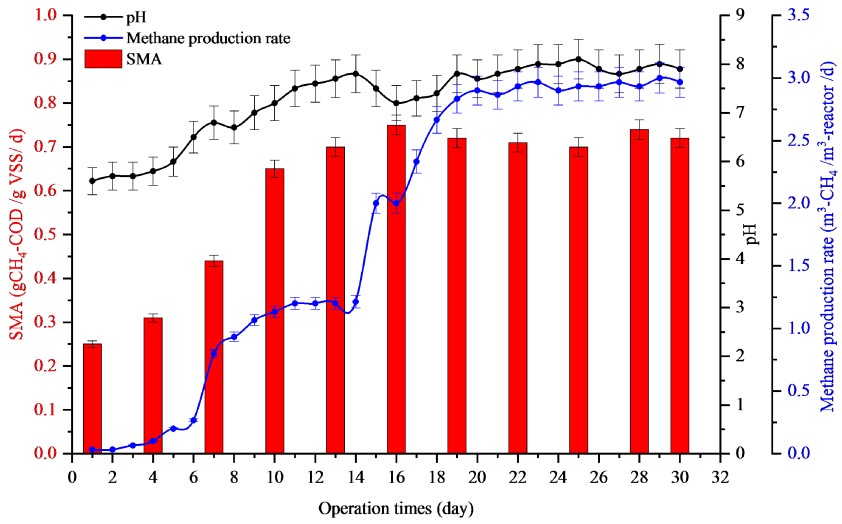

**Figure 2** Full-scale application for recovery inhibited sludge by dilution with biogas effluent at a ratio of 8:2.

and archaeal communities of these four recovered sludge were higher than those of self-recovery (Fig. 3). The bacterial community of self-recovery strategy was dominated by *Desulfotomaculum* sp., *Bacteroidetes* sp., and *Lactobacillus* sp. (Fig. 3A). The number of *Clostridium* sp., *Anaerostipes* sp., *Lynsinibacillus* sp. increased after 12 days of self-recovery. For the archaea community, self-recovery was dominated by *Methanosaeta* sp. (Fig. 3B). The numbers of *Methanosaeta* sp. and *Methanosarcina* sp. increased after 12 days of self-recovery, while the bacteria in recovery by dilution with BE, at a ratio of 8:2, was dominated by *Desulfotomaculum* sp., *Lactobacillus* sp., *Bacteroidetes* sp., *Clostridium* sp., *Staphylococcus* sp., *Selenomonas* sp., and *Lynsinibacillus* sp., in which the last four species increased after 5 days of recovery. The archaea community was dominated by *Methanosaeta* sp. and *Methanosarcina* sp. in which the latter species decreased after 5 days of recovery. The recovery by 0.14% w/v NaOH addition was dominated by *Desulfotomaculum* sp., *Blautia* sp., *Lactobacillus* sp., *Bacteroidetes* sp., and *Selenomonas* sp., with the number of *Blautia* sp., *Clostridium* sp., *Anaerostipes* sp., *Lynsinibacillus* sp., and *Staphylococcus* sp. increasing after 5 days of recovery. Furthermore, the archaeal community was dominated by *Methanosaeta* sp., *Methanosarcina* sp., and *Methanococcoides* sp. after 5 days of recovery. The recovery by 0.50% w/v Ca(OH)$_2$ addition was dominated by *Desulfotomaculum* sp., *Blautia* sp., *Clostridium* sp., *Anaerostipes* sp., *Kurthia* sp., *Exiguobacterium* sp., *Staphylococcus* sp., *Selenomonas* sp., *Bacillus* sp., *Lynsinibacillus* sp., *Lactobacillus* sp., and *Bacteroidetes* sp., in which the last two species did not appear at 5 days of recovery. The archaea community was dominated by *Methanosaeta* sp., with *Methanosarcina* sp., and *Methanococcoides* sp. at 5 days of recovery. The recovery by 8.0% w/v oil palm ash addition was dominated by *Desulfotomaculum* sp., *Blautia* sp., *Lactobacillus* sp., *Bacteroidetes* sp., *Clostridium* sp., *Anaerostipes* sp., *Kurthia* sp., *Exiguobacterium* sp., *Staphylococcus* sp., *Selenomonas* sp., *Bacillus* sp., and *Lynsinibacillus* sp., where the first four species did not appear at 5 days of

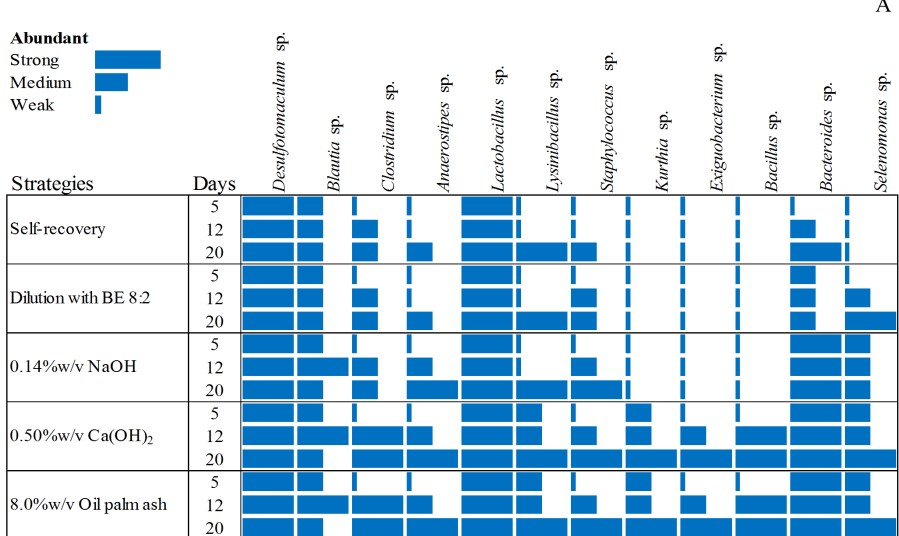

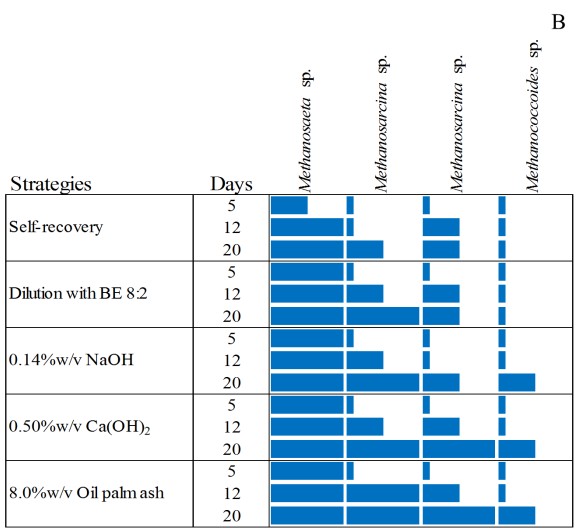

**Figure 3 Dynamic diversity of bacteria (A) and archaea (B) during recovery by self-recovery, dilution with BE 8:2, 0.14% w/v NaOH addition, 0.50% w/v Ca(OH)₂ addition, and 8.00% w/v oil palm ash addition.** The size of the rectangle respect to the dominance of microorganisms is represented from long-size to short-size for strong dominant to low dominant of microorganisms, respectively.

recovery, while the dominated archaea community is similar with the recovery by 0.50% w/v Ca(OH)₂ .The orders *Clostridiales* and *Bacilli* were observed as main bacteria in all recovered sludge. *Desulfotomaculum* sp. was found in all recovered strategies while *Blautia* sp., *Clostridium* sp., and *Anaerostipes* sp. were remarkably abundant during 12–20 days in pH adjustment with 0.50% w/v of Ca(OH)₂ and 8.0% w/v of oil palm ash addition. During the third week, *Bacillus* sp., *Lactobacillus* sp., *Staphylococcus* sp., *Kurthia* sp., *Lysinibacillus* sp., *Exiguobacterium* sp., and *Bacillus* sp. were observed in all strategies. The recovery by 0.50% w/v Ca(OH₂) and 8.00% w/v oil palm ash addition was predominant with bacteria

belonging to *Kurthia* sp., *Exiguobacterium* sp., and *Bacillus* sp. These bacteria can produce VFA from monomers after hydrolysis, especially *Exiguobacterium* sp., which was a high number in the third week. The member of *Bacteroides* sp., and *Selenomonas* sp. had a low number in self-recovery but abundant in the recovered sludge with 0.14% of NaOH, 0.50% of $Ca(OH)_2$, and 8.00% of oil palm ash addition. The distribution of the exclusive bacteria in different groups clearly showed that the recovery strategy significantly influences the bacterial community structure and selectively enriches specific acidogenic bacteria during the recovery process. The acetoclastic methanogen (*Methanosaeta* sp.) was more abundant than hydrogenotrophic methanogens in the recovered sludge. *Methanosaeta* sp. and *Methanosarcina* sp. were predominant in the recovered sludge with 0.50%w/v $Ca(OH)_2$ and 8.00% w/v oil palm ash addition. Molecular identification of bacteria and archaea from recovery strategy by dilution with BE 8:2, addition with 0.14% w/v NaOH, 0.50% w/v $Ca(OH)_2$, 8.00% w/v oil palm ash and self-recovery by denaturing gradient gel will be offered in Tables S1 and S2, respectively).

## DISCUSSION

The inhibited AD sludge was caused by low pH and high VFA accumulation, resulting in extreme alkalinity consumption from the AD reactor. The pH inside the reactor directly affected the microbial activity leading to low biodegradability and methane production. The long lag phase of self-recovery indicated that adaptation and initiating bacterial multiplication are required due to the loss of a dynamic balance between acidogens and methanogens. Methane production was observed in self-recovery under low initial pH of 5.7. Several hydrogenotrophic methanogens can grow and metabolize at acidic pH (often less than 6.0) (*Charalambous et al., 2020*). The prevalence of hydrogenotrophic methanogens over acetoclastic methanogens was also reported by Kim et al. (2004) during the bioreactor operation for hydrogen production at pH below 5.0. The dilution with tap water was not significant to accelerate the recovery of the inhibited sludge. This strategy can dilute inhibitors and the active microbes and substrates, resulting in a reduced methane production rate and providing long-time recovery. The dilution with water at the ratio of 5:5 achieved a shorter lag time (9.1 days) than self-recovery (21.2 days) but did not accelerate the recovery process (*Wu et al., 2015*). The dilution with BE at a ratio of 8:2 could enhance methane yield of 34.7% comparing with self-recovery with shorter recovery time and high $k_H$. The high $k_H$ indicates a high conversion rate of the recovered sludge (*Sosnowski et al., 2008*). The recirculation of BE to adjust the pH of POME could enhance methane production by two-stage anaerobic digestion with the highly flavored activity of acidogens (*O-Thong et al., 2016*).

The inhibited sludge was recovered by providing alkalinity sources for methane-producing bacteria (*Chen, Zhang & Wang, 2015*). It was reported that increased alkalinity resulted in higher methane production than non-alkalinity addition reactors (*Lens et al., 2003*). A significant increase in methane production (3.03%) was observed in the AD of wet poultry with NaOH addition (*Ajiboye, Lasisi & Babatola, 2018*). The adjusting pH with 0.14% w/v NaOH addition could adjust the pH of the inhibited sludge to pH 7.30. The

recovered sludge was a relatively stable pH of 6.80–7.50, enhancing the methane yield of 83.3% compared with self-recovery. The results agreed with *Zhang, Xing & Li (2018)*, who reported that the pH adjustment of the AD system by 0.013% of NaOH addition could delay the time of process failure by enhancing the tolerance of methanogens to the high concentration of VFA via reducing the ratio of un-dissociated VFA. NaOH is one of the most popular alkaline chemicals used in the AD system due to the potential to buffer the pH (*Gáspár, Kálmán & Réczey, 2007*). The recovery by Ca (OH)$_2$ addition for adjusting the pH of inhibited sludge showed high function as a buffering capacity for the AD system with a stable pH in the recovered AD systems. In addition, *Li et al. (2009)* reported the pH adjustment with Ca(OH)$_2$ improved the methanogenic activity by maintaining a stable pH for methanogens. The Ca(OH)$_2$ addition at a concentration of 0.6%w/v to 1.0% w/v could maintain active methane-producing microorganisms and stability in the AD process (*Zhang et al., 2014*). However, the accumulation of Ca$^{2+}$ may lead to the precipitation of calcium salt and accumulation on the reactor walls leading to loss of nutrition and lower buffer ability in the AD system (*Zhu, Wan & Li, 2010*). The recovery by oil palm ash addition can improve the buffer capacity and methane production rate of the inhibited sludge with a short lag phase and short recovery time. The addition of ash could increase functional buffering capacity corresponding with *Bunrung, Prasertsan & Prasertsan (2014)*, who reported that 15% (w/v) oil palm ash addition resulted in increased pH from 7.5 to 9.1. Oil palm ash composed of silicon dioxide (58–65%), calcium oxide (6–7%), and potassium oxide (7–8%) could improve buffer capacity and pH of the AD systems in the range of 8.25–9.14 (*Tangchirapat, Jaturapitakkul & Chindaprasirt, 2009*). The oil palm ash addition of 1.18% w/v into POME improved methane yield (218.79 mL-CH$_4$/g-COD) and adjusted the pH in the suitable range for AD systems (*Jijai, Muleng & Siripatana, 2017*). The recovery by NaHCO$_3$ addition had low recovery efficiency due to Na$^+$ at high concentration could inhibit the methanogens resulting in a low methane production rate (*Zhang et al., 2016*). The Na$^+$ slightly inhibited the methanogens in AD systems at 0.31% w/v of NaHCO$_3$ addition (*Chen, Cheng & Creamer, 2008*). The recovery by the addition of active methane-producing sludge can reduce the lag phase with the small improvement of biogas production comparing to self-recovery. The results in line with previous research (*Salminen & Rintala, 2002*; *Cirne et al., 2007*) that an increase of active methane-producing sludge proportion can reduce the recovery time from 45 to 28 days. Previous reports also showed that a re-inoculum size of 80% could recover the inhibition of mesophilic anaerobic sludge treating the de-oiled grease trap waste (*Wu et al., 2015*).

The high recovery efficiency strategies (0.50%w/v Ca(OH)$_2$ addition), dilution with biogas effluent at a ratio of 8:2, and 8.00% w/v oil palm ash) were dominated *Clostridium* sp., *Kurthia* sp., *Exiguobacterium* sp., *Bacteroides* sp., and *Bacillus* sp. They had been identified as being involved in biogas production, especially in hydrolysis and acidogenesis stages (*Wirth et al., 2012*). *Exiguobacterium* sp. has been confirmed as an amylase and protease producing bacterium (*Kumar et al., 2014*) and produces highly effective proteolytic enzymes (*Oh et al., 2018*). The member of *Bacteroides* sp. has been shown as a main microbe in anaerobic reactors with polysaccharide degradation (*Levén, Eriksson & Schnürer, 2007*; *Trzcinski, Ray & Stuckey, 2010*). *Desulfotomaculum* sp., *Blautia* sp., and *Clostridium* sp.

were fermentative bacteria and acetogenic bacteria that could convert soluble organics to VFAs. The *Clostridiales* was generally found in the stable AD digester (*Li et al., 2015*). The high microbial diversity in recovered sludge results in higher microbial functions with more stability in operation and good AD performances (*Carballa, Regueiro & Lema, 2015*). The acetoclastic methanogens were more abundant than hydrogenotrophic methanogens in the recovered AD system. The inhibited sludge commonly induced hydrogen production and consequently facilitating the growth of hydrogenotrophic methanogens (*Liu et al., 2016*). The hydrogenotrophic methanogens were decreased in the recovered AD system. The recovery by 0.50% w/v $Ca(OH)_2$ and 8.00% w/v oil palm ash addition enhanced acetoclastic methanogens resulting in the highest methane yield and methane production rate. The dominant *Methanosarcina* sp. was most important in the recovered AD system, which utilization acetate to produce $CH_4$. Maintenance the number of *Methanosarcina* sp. during the AD process is critical for the stability of performance (*Yang et al., 2016*).

## CONCLUSIONS

Recovery of the inhibited sludge by addition of 0.14% w/v NaOH, 0.50% w/v $Ca(OH)_2$, 8.00% w/v oil palm ash, and dilution with biogas effluent at a ratio of 8:2 had a short lag phase with a short recovery time of 30–36 days. The dilution with biogas effluent at a ratio of 8:2, 0.14% w/v NaOH addition, and 8.00% w/v oil palm ash addition was considered a more economical strategy with a recovery time of 30-36 days. The recovered AD system can increase methane yield by 35–83% and significantly higher kinetics, SMA activity, and short lag phase comparing to self-recovery. The *Clostridiales* sp., *Bacilli* sp., and *Methanosarcina* sp. were dominated in the recovered AD system. The imbalanced full-scale hybrid cover lagoon reactor (6,000 $m^3$) was recovered within 15 days by dilution with biogas effluent at a ratio of 8:2 with a better result from the lab-scale (36.4 days).

### Funding

This study was financially supported by the Thailand Research Fund through a Ph.D. scholarship to Nantharat Wongfaed through the Royal Golden Jubilee Ph.D. (Grant No. PHD/0108/2556), and Senior Research Scholar (Grant No. RTA6080010). The funders had no role in study design, data collection and analysis, decision to publish, or preparation of the manuscript.

### Grant Disclosures

The following grant information was disclosed by the authors:
Thailand Research Fund:  PHD/0108/2556.
Senior Research Scholar:  RTA6080010.

### Competing Interests

The authors declare there are no competing interests.

## Author Contributions

- Nantharat Wongfaed and Sompong O-Thong conceived and designed the experiments, performed the experiments, analyzed the data, prepared figures and/or tables, authored or reviewed drafts of the paper, and approved the final draft.
- Prawit Kongjan conceived and designed the experiments, performed the experiments, prepared figures and/or tables, and approved the final draft.
- Wantanasak Suksong conceived and designed the experiments, analyzed the data, authored or reviewed drafts of the paper, and approved the final draft.
- Poonsuk Prasertsan conceived and designed the experiments, performed the experiments, authored or reviewed drafts of the paper, and approved the final draft.

## Data Availability

The raw measurements are available in the Supplementary Files.

## Supplemental Information

Supplemental information for this article can be found online at http://dx.doi.org/10.7717/peerj.10592#supplemental-information.

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
