# Peer review of "Strategies for recovery of imbalanced full-scale biogas reactor feeding with palm oil mill effluent"

_PeerJ, doi:10.7717/peerj.10592_

## Round 0.1 · original submission · Major Revisions

Thank you for your submission. I would like to let you know that your manuscript can be recommended for publication provided you are able to address the major revisions according to the attached referee reports. Like our referees, I also would like to emphasize the need for a detailed grammar check by a native English speaker and strong improvements on the validation of the findings with comprehensive discussions and conclusions. I also can suggest for re-selecting the keywords as some of them cover very general topics/terms, Once again, thank you for submitting your manuscript to PeerJ and I look forward to receiving your revision.

Reviewer 1 ·

Basic reporting

Although I am not a native English speaking I consider that the manuscript should be carefully checked by a native English speaker.

The number of references (61) is a little bit high for this kind of articles, although they are adequate to the manuscript topic. However less than 15% are really recent (three last years). The authors could include some more recent references.

Article structure is adequate, and figures and tables supply the required information.

Experimental design

The manuscript topic is within the Aims and scope of PeerJ.

The manuscript clearly defines the research topic and the applied methodology could be adequate although it presents no more than a first step. Experimental design is carried out in batch test and with no more that one feed; this implies that there is no a description of what will occur in a long term test (for example a second feed).

Methodology description is adequate, although there are some specific item as:
- The determination of first-order kinetic should be explained with more detail (Was the lag phase considered?)
- Check equation 1. What is "ln"?

Validity of the findings

The main concern of the manuscript is the limited validity of the data as they were obtained in batch test and single test (as is indicated in previous section).

Discussion should be improved as there is no a real comparison among the used strategies. It looks like a mere data exposition, without a real discussion and clear conclusions.

Some specific items are:
- The self-recovery sludge has very high methane yield and methane production rate. How can this fact be explained if the test was carried out a pH lower that 5.6?
- Economic evaluation should include a clear comparison of the different strategies.
- Figure 1. How can be explained that the Acetic acid SMA is lower for the original microbial sludge than for the inhibited sludge? These data should be discussed in the manuscript.

Reviewer 2 ·

Basic reporting

The English and writing style needs to be improved. There are many sentences need to be rephrased. For example:

Line 434-436, Page 19: The pH adjustment with Ca(OH)2 and dilution with 20% v/v of biogas effluent can recovery the acidified and low pH inhibition of POME feeding commercial biogas plant with economically feasible.

Figure 1: There are no error bars in Figures 1. In order to assess the significance of any differences, it is essential to conduct at least duplicate experiments, to provide error bars, and to provide an assessment of statistical significance.

Proper reference should be given for the following statements.

Line 258-260, Page 14: ‘It has been reported that the addition of NaHCO3 resulted in a relatively high level of alkalinity and pH in the digester, which gradually causes specific inhibition on the metabolism and growth of methanogens.’

Line 397-398, ‘The oil palm ash composed of silicon dioxide (58-65%), calcium
oxide (6-7%), and potassium oxide (7-8%)…’

Line 174: Please check the referencing style

Experimental design

No comment

Validity of the findings

Page 14: The hydrolysis constants (kh) in Table 3 should be discussed in the context.

Results and discussion:
This section should be restructured, as there are some repetitive discussion of the results in Page 12-19, especially in the subsection of discussion (page 17-19).

Additional comments

This paper describes the strategies for recovering inhibited biogas reactor feeding with palm oil mill effluent. Overall, the discussion of the results is simple; the authors should provide more justifications for each major finding with references. The manuscript can be published; nevertheless the English, sentence structure and writing style in most of the parts not up to the mark and should be checked thoroughly. Please state clearly the area of improvement/future research area in the conclusion.

---

## Round 0.2 · Major Revisions

Thank you for your re-submission and your efforts to fulfill our referees’ previous comments. I would like to recommend moderate to major revisions for the publication of your manuscript in PeerJ. I mainly suggest fulfilling the additional comments of our first referee on improving the discussion and the description of several experimental data in a more detailed and clear way (as specifically indicated within the referee’s comments). I also think that the conclusions and discussions sections needs further improvements than their current states. I look forward to receiving the new improved version of your manuscript.

Reviewer 1 ·

Basic reporting

Article structure is adequate, and figures and tables supply the required information.
The manuscript has been modified according to reviewers´ comments, but there are still some points that could be improved.

Experimental design

Methodology description is adequate, although there are some issues to be considered as is indicated in comments to author.

Validity of the findings

Discussion should be improved to compare the used strategies based the experimental data.

Additional comments

Some general comments:

- Is the full-scale hybrid cover lagoon reactor (6,000 m3) located in the Prasang Green Power Co., Ltd. palm oil mill? The description of the strategy applied to the full-scale reactor (lines 148 – 151) is not clear enough. The inhibited sludge is sludge from the same reactor? I suppose that the biogas effluent is the effluent from another reactor, but is not clearly indicated.
- Table 4. Each item should be described. For example, what is the meaning of biogas loss (there is also a biogas production) and why is different for each recovery strategy?
- Recovery time is not clearly defined in the text.
- Specific methanogenic activity (SMA) test should be described and discussed more deeply. It is not easy to understand that SMA with acetic acid would be lower than SMA with cellulose as for inhibited sludge and 0,50% w/v Ca(OH)2. The meaning of the different SMA should be discussed.
- The recovery of AD process is based on methane yield, methane production rate and SMA, but there is no a clear discussion of the meaning of each one of these parameters. The highest methane yield should be not very different for each test, except if there were a clear inhibition. The ratio between the highest methane production rate for each test and SMA will be the same if the SVadded is the same for each test; if SVadded is not the same for each test, the comparison between the highest methane production rate does not supply any useful information.

Reviewer 2 ·

Basic reporting

The manuscript has been improved to an acceptable level.

Experimental design

No comment

Validity of the findings

No comment. Amendments have been done accordingly.

Additional comments

No comment. Amendments have been done accordingly.

---

## Round 0.3 · accepted · Accept

Thank you for your second re-submission and your additional efforts to fulfill our referees’ further comments. I am pleased to inform you that your manuscript has been accepted for publication in PeerJ. I also would like to thank you for your contribution. Best wishes.
Ela